# Smaller climatic niche shifts in invasive than non-invasive alien ant species

Olivia K. Bates [ID] [1✉], Sébastien Ollier[2] & Cleo Bertelsmeier [ID] [1✉]

The globalization of trade and human movement has resulted in the accidental dispersal of thousands of alien species worldwide at an unprecedented scale. Some of these species are considered invasive because of their extensive spatial spread or negative impacts on native biodiversity. Explaining which alien species become invasive is a major challenge of invasion biology, and it is often assumed that invasiveness is linked to a greater ability to establish in novel climates. To test whether invasive species have expanded more into novel climates than non-invasive alien species, we quantified niche shifts of 82 ant species. Surprisingly, invasive species showed smaller niche shifts than non-invasive alien species. Independent of their invasiveness, the species with the smallest native niches and range sizes, experienced the greatest niche shifts. Overall, our results challenge the assumption that invasive species are particularly good pioneers of novel climates.

[1] Department of Ecology and Evolution, Biophore, UNIL-Sorge, University of Lausanne, 1015 Lausanne, Switzerland. [2] Université Paris-Saclay, CNRS, AgroParisTech, Ecologie Systématique Evolution, 91405 Orsay, France. ✉email: olivia.bates@unil.ch; cleo.bertelsmeier@unil.ch

Ever-increasing trade and travel have facilitated unprecedented globalization and the dispersal of an increasing number of species outside of their native ranges[1,2]. This has resulted in over sixteen-thousand 'alien' species globally[3]. Some of these species are considered 'invasive' as they cause significant detrimental impacts on ecosystems, human health, and economies worldwide[4,5] (See Table 1). A major challenge is to understand why only some and not all alien species become invasive[6]. The identification of ecological characteristics distinguishing non-invasive alien and invasive species, which would allow predictions of future invasions, is still considered as the holy grail of invasion biology[6,7]. It has been hypothesized that traits conferring high plasticity and adaptability to new conditions may favor establishment and spread in new environments, thereby increasing invasiveness[8–10]. For example, previous research has suggested that invasive species show higher plasticity under new environmental conditions[11], a greater ability to maintain dense monocultures[12], and higher levels of allelopathy[13]. Such traits could be particularly important when a species is introduced to areas where the climate is different from its native range. Yet, it is still unknown whether the ability to colonize novel climates contributes to the success of invasive species[14].

Although many studies have compared realized climatic niches within native and non-native ranges, the majority focus on such "niche shifts" in a single invasive species, with few including multiple species (but see e.g., refs. [15,16]). Importantly, no study has evaluated whether climatic niche shifts of invasive species are in fact larger than those of non-invasive alien species, a key prediction of plasticity-based hypotheses of invasiveness. To test this prediction, we focused on ants (Formicidae), a taxon which includes particularly prominent alien and invasive species[17]. Ants are highly successful due to their complex social structure and variety of lifestyles, behaviors, diet requirements and nest constructions[17,18]. Importantly, ants have been dispersed accidentally[19] and there has been no human selection for 'hardy' traits, such as increased tolerance to climatic conditions, as in other taxa such as plants. Moreover, they are present across all major terrestrial habitats and thrive under a wide range of climatic conditions[20] and previous work on Solenopsis invicta has suggested that ants may be able to colonize novel climates not present in their native range[21]. There are currently more than 200 known alien ant species, with 19 classified as 'invasive' by the IUCN due to their impacts on biodiversity, ecosystem functioning, agriculture, infrastructure and human or animal health, causing important economic losses[17,22].

Here, we quantified the frequency and extent of niche shifts in 82 ant species. We show that invasive species displayed smaller niche shifts than non-invasive alien species. Instead, native range and niche size impacted the extent of niche shifts. Our results challenge the hypothesis that invasive species are better at colonizing novel climates, and has implications for predictive species distribution modeling.

## Results and discussion

We assessed niche shifts of all 82 alien and invasive ant species for which at least 10 occurrence points in both their native and non-native ranges were available (median number of occurrence points per species: 209, range: 27–3531. Importantly, the number of occurrences per species had no effect on the results (See "Methods" section). We quantified different aspects of niche shifts between native and non-native niches (Table 1) by calculating niche overlap (Schoener's D; shift in niche centroid, range: 0–0.81) and expansion into novel climates (percent of non-native niche extending beyond native niche, range: 0–100%). We also performed niche equivalency tests (probability of observed niche shift due to chance) which revealed that all but two of the 82 species, Nylanderia bourbonica ($p = 0.08$) and Tetramorium bicarinatum ($p = 0.07$), had significantly divergent niches between the native and non-native range, compared to random ($p < 0.05$) (see Supplementary Data 1). Some species had high niche overlap and low expansion between their native and non-native range (Fig. 1a), whereas others showed no niche overlap and high expansion (Fig. 1b), indicating near complete niche shifts. However, low niche overlap does not necessarily equate to high expansion. For example, some species had low niche overlap and low expansion (Fig. 1c), with limited non-native niches largely encompassed by the native niches. Other species had high niche overlap and high expansion (Fig. 1d), representing non-native niches that largely include the native niche, but also extend beyond it.

Surprisingly, invasive species had on average higher niche overlap with their native range than non-invasive alien species

**Table 1 A glossary of key terms used in the study, split between definitions of keywords and concepts used, and metrics used in measurements of niche change.**

| Term | Definition |
| --- | --- |
| Keywords | |
| Alien species | A species that has been introduced via human-mediated dispersal (accidental or intentional) to an area outside of its native range, where it has established a self-sustaining population. |
| Invasive Species | An alien species with detrimental impacts on native biodiversity, health or the economy, following the definition by the invasive species specialist group (ISSG) of the IUCN. |
| Native range | The natural geographic distribution of a species without human intervention. |
| Niche shift (climatic) | The newly established outdoors range of an alien species. |
| Niche expansion (climatic) | Establishment of a population in climatic conditions outside of the native realized niche of the species. |
| Fundamental niche | The full set of environmental conditions under which a species could thrive in the absence of competition or dispersal constraints. |
| Realized niche | The environmental conditions under which a species lives as a result of limiting factors (typically a subset of the fundamental niche). |
| Niche metrics | |
| D overlap (Schoener's D) | A measure of niche overlap. The overlap of the density of occurrences between two populations in niche space, ranging from 0 (no overlap) to 1 (complete overlap)[60]. |
| Equivalency test | A permutation test where all occurrences are pooled and randomly split, to simulate a random distribution of D overlap values. If 95% of the simulated overlap is higher than the observed D value, the assumption of niche equivalency can be rejected[60]. |
| Expansion | The percentage of the non-native niche that is not present in the native niche[16]. |

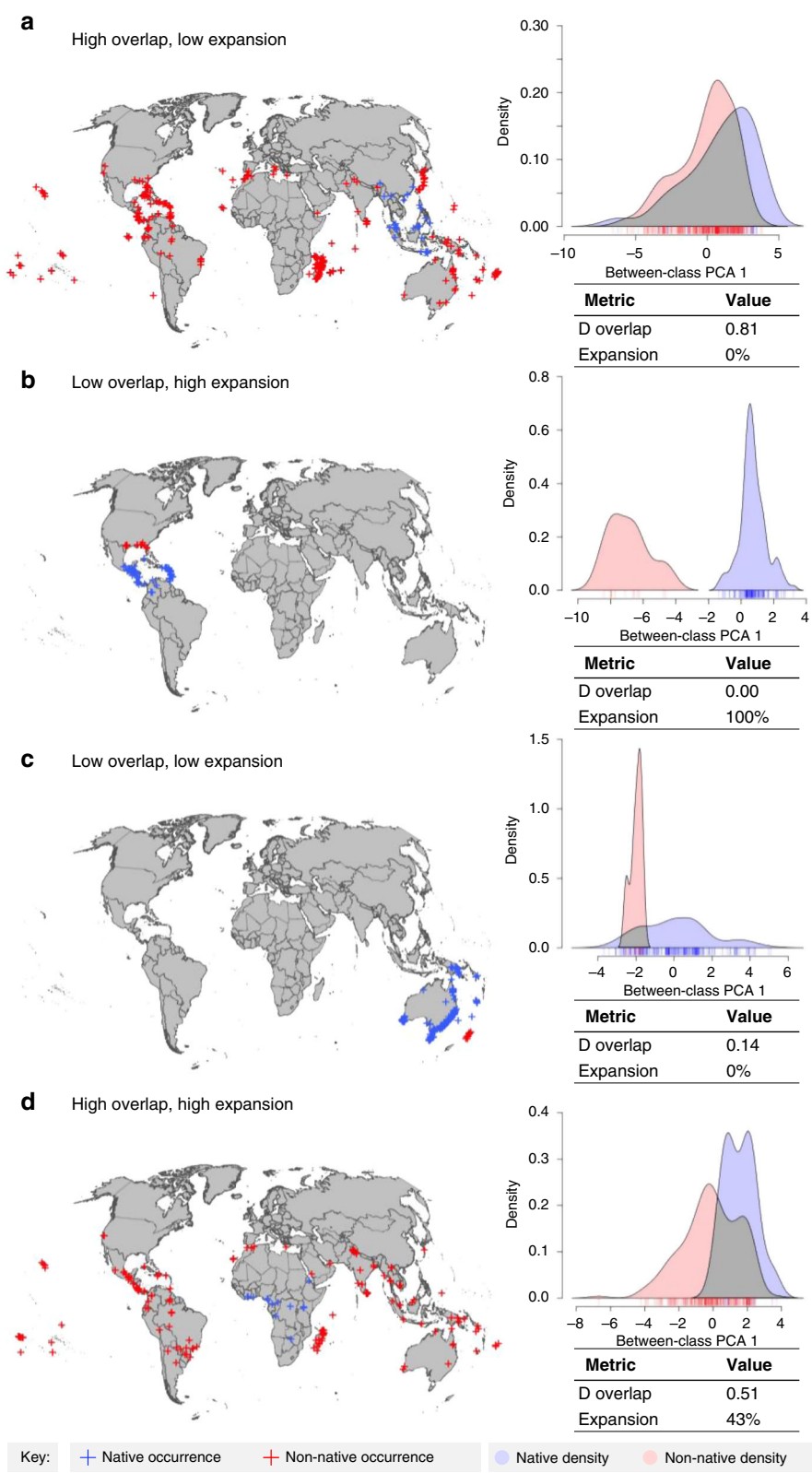

**Fig. 1 Shifts in climatic niche and geographic distribution between native and non-native ranges.** Species demonstrated are: (**a**) *Tetramorium bicarinatum*, (**b**) *Strumigenys margaritae*, (**c**) *Amblyopone australis*, and (**d**) *Monomorium pharaonis*.

(Kruskal–Wallis sum rank test, $X^2 = 4.19$, df = 1, $p = 0.04$), and the amount of niche expansion did not differ between non-invasive alien and invasive species (Pearson's chi-squared test, $X^2 = 1.50$, df = 1, $p = 0.22$) (Fig. 2b). That is, while the average expansion into novel climates was not different between invasive

and alien species, invasive species tended to have less divergent niche centroids between their native and non-native niches. These results demonstrate that invasiveness in ants, defined by the severity of ecological impacts, does not rely on a greater propensity for climatic niche shifts. Indeed, invasive species

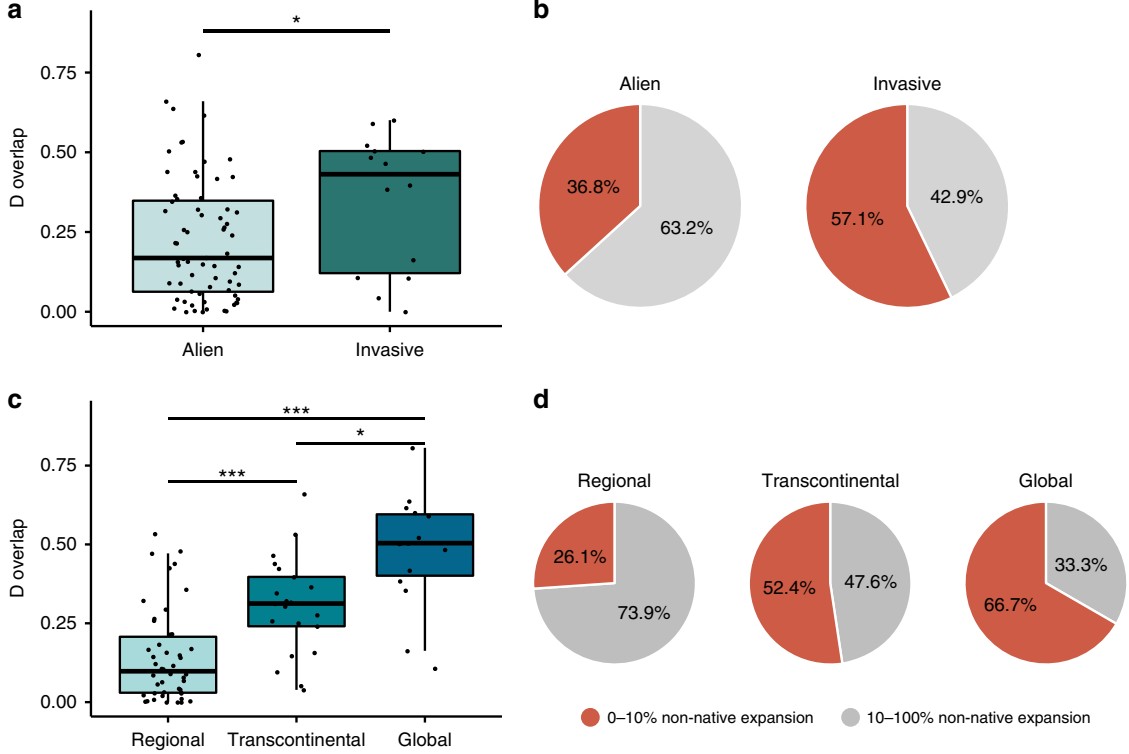

**Fig. 2 Comparison of niche similarity between the native and non-native ranges of different ant species using different definitions of invasiveness.**
**a** Compared to non-invasive alien species ($n = 68$ species), IUCN-classified invasive species ($n = 14$ species) have a larger D-overlap (Kruskal–Wallis sum rank test, $p = 0.04$) and (**b**) the similar percentages of species that have expanded above 10% (Pearson's chi-squared test). **c** Between species distributed at regional ($n = 46$ species), transcontinental ($n = 21$ species), and global ($n = 15$ species) levels, with increasing levels of global dispersion quantifying higher invasiveness, there are increasing amounts of D-overlap (Post-hoc Dunn test with Benjamini-Hochberg correction, regional-transcontinental, $p < 0.001$; transcontinental-global, $p = 0.02$; regional-transcontinental, $p < 0.001$), and (**d**) decreasing percentage of species with expansion above 10%, as levels of geographical dispersion increase (regional to transcontinental to global, Pearson's chi-squared test). Boxplots elements show; center line, median; box limits, upper and lower quartiles; whiskers, 1.5× interquartile range.

showed overall smaller niche shifts compared to non-invasive alien species.

Because invasiveness can also be defined in terms of total spatial spread instead of impacts, with higher spatial spread relating to higher invasiveness, we tested if more widespread species have a greater propensity for climatic niche shifts. To do this, we separated the 82 species into three groups of similar levels of geographic dispersion (following[23]). For each species, we determined the number of political entities within which it had established (spatial richness, see "Methods" section) and estimated the Rao spatial diversity of its entire range, taking into account the pairwise distances between the centroids of these regions[23]. Using a cluster analysis within this richness-diversity space, species were classified as members of a regional, transcontinental, or global dispersion group (see "Methods" section,[23]). Species in the regional dispersion group mostly occur across multiple countries within the same continent. Transcontinental species have spread across multiple continents but only to a few countries within each. Global species have successfully dispersed across continents and spread throughout the countries within each continent. Species that were classified as invasive by the IUCN based on their impacts occurred in all three dispersion groups (global = 9, transcontinental = 2, regional = 3). Interestingly, species with a broader total geographical distribution exhibited a higher $D$ overlap (Kruskal–Wallis test, $X^2 = 31.04$, df = 2, $p < 0.001$) (Fig. 2c). Moreover, niche expansion decreased with increasing total geographical range (Pearson's chi-squared test, $X^2 = 10.80$, df = 2, $p < 0.001$) (Fig. 2d). This demonstrates

that invasiveness, defined as greater spatial spread, is also associated with smaller climatic niche shifts.

We acknowledge that some non-invasive alien species may still be in the early stages of their invasion process and will become invasive in the future[24], due to time lags between initial introduction and subsequent spread or impacts of the species[25,26]. However, this is likely to concern only a few species, given that most alien ant species have been moved by humans around the planet for a long time. The vast majority of currently alien ant species has indeed started colonizing new areas even before World War II[23]. Therefore, most species currently recorded as "alien" have likely had sufficient time to become invasive.

It is counterintuitive that for both definitions of invasiveness, species with greater invasiveness exhibit smaller climatic niche shifts. This may be because the colonization of novel climates has nothing to do with a species' propensity to spread or cause impacts on native ecosystems. Species distributions are known to be the result of several limiting factors, as summarized by the BAM (Biotic, Abiotic, Movement) model[27]. This conceptual scheme, which has been widely used in invasion ecology, displays areas with the necessary biotic interactions (B), the areas with necessary abiotic conditions (A) and areas that can be reached by the species via movements in space (M). A species is able to thrive in places where all three conditions are met. Our results suggest that there is greater overlap between the B, A, and M areas for invasive than non-invasive species. For alien species, there might be large areas which meet the abiotic conditions for its survival (i.e., the species fundamental niche), but cannot be colonized by

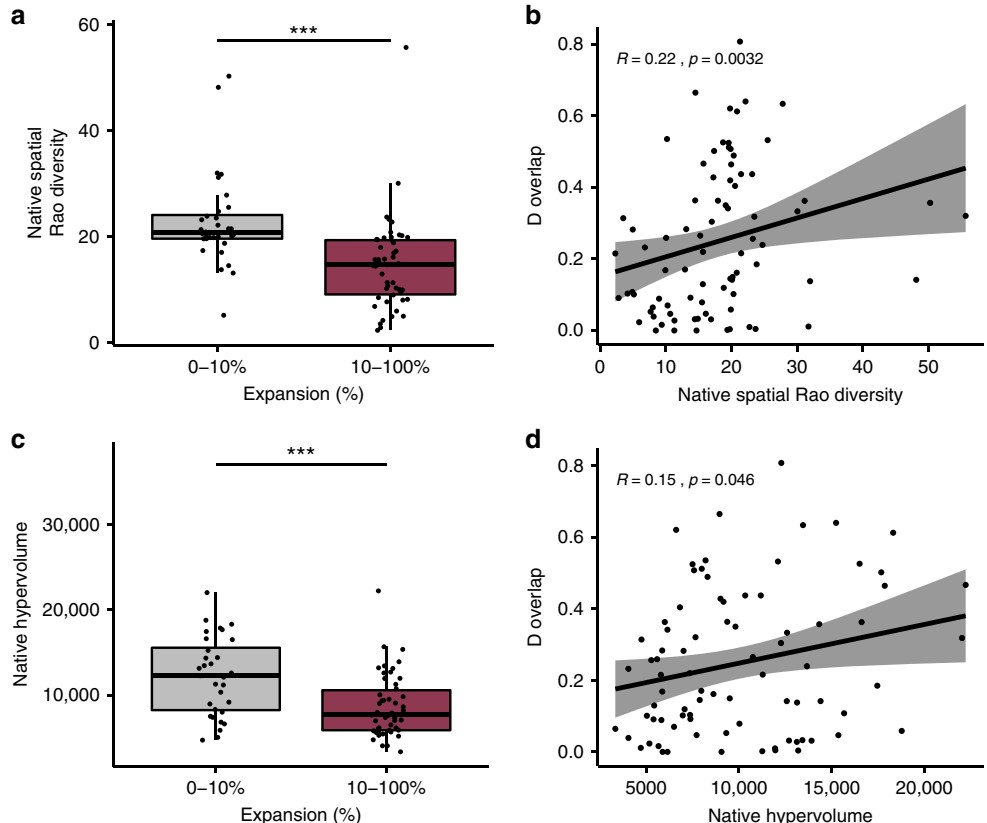

**Fig. 3 Effects of native range characteristics on the level of expansion and *D* overlap between native and non-native ranges on different ant species.**
**a** Species with low expansion (0–10%) (*n* = 32 species) had a higher native spatial Rao diversity than species that expanded above 10% (*n* = 50 species), and (**b**) increasing native spatial Rao diversity also led to higher *D* overlap (*n* = 82 species) (Kendall's Tau rank correlation coefficient test, *p* < 0.001).
**c** There was a higher Native Hypervolume size for the low expansion group, and (**d**) with increasing Native Hypervolume size, *D* overlap also increased (Kendall's Tau rank correlation coefficient test, *p* < 0.001). Kendall Rank Coefficient regression lines are observed with 95% confidence intervals. Boxplots elements show; center line, median; box limits, upper and lower quartiles; whiskers, 1.5× interquartile range.

the species due to biotic interactions or dispersal barriers[28,29]. Establishment outside of their native range would therefore more frequently result in observed "niche shifts", when these species are released from B and M constraints (e.g. refs. [30,31]).

To test whether niche shifts are more frequent in species with a smaller native range, we calculated Rao spatial diversity for the native range (as a measure of native geographic dispersion) for each species. We found that species which showed minimal niche expansion (<10%)[16,32], were more widely dispersed in their native range (Rao spatial diversity, Kruskal–Wallis test, $X^2 = 22.86$, df = 2, *p* < 0.001) (Fig. 3a). Higher Rao spatial diversity also correlated to higher *D* overlap (Kendall Rank Coefficient, rτ = 0.22, *p* = 0.003) (Fig. 3b). Because the extent of spatial spread does not always represent the diversity of climatic conditions within that range, we also assessed the size of the species' climatic niche within its native range. To do that, we calculated the n-dimensional niche hypervolume of the species' native range[33], using multidimensional kernel-estimation in PCA space (see "Methods" section). Species that showed little expansion also had significantly larger niche hypervolume in their native range (Kruskal–Wallis test, chi-squared = 12.17, df = 2, *p* < 0.001) (Fig. 3c), furthermore, species with higher *D* overlap had larger native niche hypervolumes (Kendall Rank Coefficient, rτ = 0.15, *p* = 0.05) (Fig. 3d).

Native range size and niche hypervolume were independent of impact-based invasiveness (native Rao spatial diversity: Kruskal–Wallis test, $X^2 = 0.22$, df = 1, *p* = 0.64, native hypervolume: Kruskal–Wallis test, $X^2 = 0.32$, df = 1, *p* = 0.57). However, native range size was linked to dispersion-based invasiveness

(Kruskal–Wallis test, $X^2 = 8.32$, df = 2, *p* = 0.02), indicating that species that are widespread within their native range also realized the largest global distributions. Despite this, the native niche hypervolume was also independent of dispersion-based invasiveness (Kruskal–Wallis test, $X^2 = 1.61$, df = 2, *p* = 0.45). Therefore, the size of the native niche volume conditions the extent of niche shifts independently of the species' invasiveness. Species with smaller niche shifts may have had less constrained realized niches in the native range. On the contrary, species with a low natural dispersal capacity and small niche size in their native range may benefit more from human-mediated dispersal, allowing them access novel climatic conditions within their fundamental niche[34].

The negative relationship between the size of the native niche hypervolume or geographic range, and the extent of niche shifts suggests that it is generally the realized niche rather than the fundamental niche which has shifted. For example, the observed niche shifts could be explained by increases in thermal plasticity in the invaded range, given that critical thermal limits may change with season[35] and diet[36] in ants. Realized niche shifts can occur because of relaxed ecological constraints, and do so in the absence of adaptive evolution in the non-native population[37–39]. However ruling out that evolution, either through founder effects[40], novel genetic combinations[41] or adaptive evolution[42], has altered climatic tolerance during the invasion process of individual species would require experimental evidence[42–44]. Given the high frequency of niche shifts found in our study, we hope that future research will disentangle potential mechanisms explaining these shifts by combining computational and

experimental approaches. To better understand the dynamics of niche shifts, it would also be interesting to quantify the expansion of the non-native niche over time for species with documented invasion histories. This would allow testing if species start by colonizing climatically similar areas before shifting their niche or, if on the contrary they tend to colonize novel climates first before filling their niche when they arrive in their introduced range. Previous research found that the highly ant species *Solenopsis invicta* tended to first invade areas more similar to the native range[21], but it is not yet known if alien species in general follow such dynamics.

Our findings have implications for predictive species distribution models (SDMs). These models make the strong assumption of niche conservancy between native and non-native ranges. Many studies have used SDMs to predict potentially suitable areas for invasive species[45]. However, if niche shifts occur during invasion, be they realized or fundamental niche shifts, these SDM predictions will not be reliable[46]. But for invasion biologists, it is encouraging that the most invasive species showed the smallest niche shifts, indicating that SDM predictions or climate-matching risk assessments for species which pose the greatest environmental risks may be the most reliable. However, many non-invasive alien ant species were capable of colonizing novel climates not present in the native range. This suggests that studies assuming niche conservancy cannot predict accurately the future threat of these species, particularly those with small native niche sizes. More generally, this is not just a problem for modeling potentially suitable areas of alien species, as SDMs are a major tool used to predict the effects of climate change for endemic and threatened species[47], which typically have much smaller geographical ranges than non-invasive alien and invasive species. In order to predict a species' capacity to expand its climatic niche, experimental evidence is needed. Incorporating experimental data on changes in physiology, phenology or species interactions in response to environmental changes into more mechanistic models may capture a more realistic view of the climates that can be tolerated (e.g., refs. 48–50).

Overall, our findings reveal that, contrary to expectations, invasiveness is not linked to a species' ability to shift its niche. This challenges the assumption that invasive species are particularly good pioneers of novel climates. Instead, we found that the characteristics of a species' native range were linked to the ability to colonize novel climates. Species with small ranges and niches in their native range showed larger niche shifts. These findings caution against using SDMs to predict future invasions of species with small geographic distributions in their native range since they may be constrained by other ecological factors and therefore not be representative of the full range of conditions under which the species can thrive.

## Methods
**Distribution data**. We compiled distribution data for all 241 ant species known to have expanded beyond their native range, using the authoritative database AntMaps[51,52]. Native and non-native ranges were distinguished in AntMaps based on the published literature. We cleaned all occurrence records by removing any dubious or indoor occurrences from the analysis. To account for sampling bias, we used the nearest neighbor distance (NND) method to thin the data, where occurrence points that were ≤0.02 units away from each other were removed (roughly 2 km) to avoid errors due to spatial autocorrelation[53]. As the resolution of the climate maps was larger than this distance, duplicate records in the same climate grid cells were removed. In addition, species that had fewer than 10 occurrence points in their native or non-native range were also removed from the analysis. Therefore, in total 82 ant species were used for the analysis. Invasive species were defined according to the categorization of the Invasive Specialist Group (ISSG) of the IUCN[54]. All other species with non-native ranges were classified as non-invasive alien species.

Using the AntMaps classification, the world map was sectioned into polygons covering all landmasses. For larger countries, such as the US, these polygons were on a provincial/state level, while for smaller countries they were on a country level[51]. Niche shift levels were compared between species grouped into 'regional',

'transcontinental', and 'global' dispersion categories. Species were assigned to these dispersion groups according to the number of polygons occupied and the geographical distance between occupied polygons, using the methods of Bertelsmeier et al.[23]. Polygons were used to account for the varying sizes of politically-defined countries.

**Climate data**. Current global climate data was sourced from the WorldClim Global Climate Database at a resolution of 2.5 arc-minutes[55]. For each occurrence point of all 82 species, climate data was extracted using 17 of the 19 available bioclimatic variables. We excluded BIO2 and BIO7 in our analysis because these values are derived from a combination of other bioclimatic variables.

**Niche shift analysis**. For each species, the climatic variables for each occurrence point in both the native and non-native ranges were reduced using PCA with ade4 package[56]. We performed a between-class analysis using the 10 axes of the resulting PCA using 'native' and 'non-native' as a priori classes to identify the axis separating these two ranges the most for each species[15]. For all three niche shift metrics, we used this axis to define the climatic niche. Using the methods of Broennimann et al.[57], this axis was rescaled into $100 \times 100$ grid cells and converted into densities of occurrences using the R package 'ecospat'[58]. The occurrences for each range were then smoothed using kernel density smoothers, to control for errors in sampling efforts. This allowed us to directly compare the niches in the native and non-native ranges in environmental space, while considering all available climates.

We determined the intersection of the occurrence densities in environmental space using Schoener's D (D)[59,60]. This measure of niche overlap between the native and non-native ranges produces a value between 0 (no overlap) and 1 (complete overlap).

We defined niche expansion as the percentage of the non-native range that is not present in the native range. For this study, expansion was categorized into non-significant (<10%) and significant expansion (>10%)—as this threshold has previously been used for classification in a significant proportion of studies[16,32]

Finally, we also performed niche equivalency tests[57,60]. All occurrences were pooled ($N–E_{pool}$) and randomly split into two datasets, at the same observed ratio of native to non-native occurrences for each species, then the D overlap was calculated. This process was repeated 1000 times. The distribution of the simulated overlaps was then compared with that of the observed D value. If the D value was lower than 95% of the simulated values, the hypothesis of niche equivalency was rejected. We adjusted the p-values for multiple statistical comparisons using Benjamini–Hochberg correction[61].

**Native range and niche size**. Geographic dispersal within the native range was calculated by determining the pairwise geographical distance between the centroids of the occupied polygons of each species' native range, from which a dissimilarity matrix was constructed. Then, Rao's quadratic entropy was calculated for each species' native dispersal, to provide a 'Rao spatial diversity' value[23]. Native niche size was calculated from the volume of the native range n-dimensional niche using the R package 'hypervolume'[33]. This method allows high-dimensional estimation of the niche using multidimensional kernel density estimation to calculate the density distribution of species records. This was calculated from PCA-space of all climates on earth derived from the 17 bioclimatic variables, which allowed comparable estimations of niche volumes between different species. For each species, the native occurrences were projected into hypervolume space using the gaussian method with a chunk size of 500. Bandwidths were fixed for every species, calculated as the maximum bandwidth when hypervolumes were calculated preliminarily using the maximum bandwidth for each axis derived from the 'free_bandwidth' option within the R package 'hypervolume'[33].

**Statistical tests**. The differences in the D overlap between the non-invasive alien and invasive groups were compared using a Kruskal–Wallis test. The differences in the level of expansion (<10% vs. >10% expansion) between the non-invasive alien and invasive groups, was compared using chi-squared tests. We also tested if species with different levels of dispersion differed in D overlap using a Kruskal–Wallis test, and observed pairwise differences using a post-hoc Dunn test with Benjamini–Hochberg correction for multiple comparisons. Between dispersion groups, the levels of expansion were compared using chi-squared tests. To compare the level of expansion to native range dispersion and niche size, the different levels of expansion were compared for both native hypervolume size and native Rao diversity using a Kruskal–Wallis test. Correlations between D overlap and both native hypervolume size and native Rao diversity were tested with Kendall's Tau rank correlation coefficient test.

We tested if the number of occurrence points had an effect on the results. To do this, we tested if the test metrics D overlap and Expansion changed with the number of native occurrence points (after thinning- see "Methods" section). We found no correlation between D overlap and the number of native occurrence points (Kendall's rank correlation tau, Rτ = −0.04, p = 0.63). There was also no difference in the number of occurrence points among species belonging to different expansion groups (10–100%), (Kruskal–Wallis test, $X^2 = 2.65$, p-value = 0.10).

**Reporting summary**. Further information on research design is available in the Nature Research Reporting Summary linked to this article.

## Data availability

All data used in this study can be downloaded from Github (https://github.com/OliviaKBates/AlienInvasiveNicheShift/blob/master/data_invasiveants.RData)[62]. Worldclim Global Climate Database (https://www.worldclim.org) and the AntMaps database (https://antmaps.org) can both be accessed through their respective websites.

## Code availability

All analyses were done in R version 3.6.0[63] and a script is supplied to generate all figures using the R workspace which contains all data used is available on Github (https://github.com/OliviaKBates/AlienInvasiveNicheShift)[62].

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

## Acknowledgements

We thank Timothy Szewczyk for comments on an earlier version of the manuscript. C.B. and O.K.B. were supported by funding from the canton Vaud and grants from the Swiss National Science Foundation and the *Programme de la Famille Sandoz–Monique de Meuron pour la relève universitaire*. S.O. and C.B. were supported by an International Emerging Action entitled "Globalization and insect invasions (GLOBINV)" and funded by the CNRS.

## Author contributions

O.K.B., S.O., and C.B. designed the research. O.K.B. performed the research, and O.K.B. and S.O. analyzed the data. O.K.B., S.O., and C.B. all contributed to the writing of the paper.

## Competing interests

The authors declare no competing interests.
