## [Peer Review File · Nature Communications]

Reviewers' Comments:

Reviewer #1:

Remarks to the Author:

The authors ask if invasive species' success stems from being able to colonise novel climates. Specifically, they test if non-native populations of invasive species exhibit greater climate variation relative to native populations, than alien species (that is, species that have successfully introduced but are not considered "invasive"). They find that introduced population of "invasive" ants are, on average, closer to the climate range of native populations than non-invasive alien ants. This finding challenges the prediction that increased plasticity or ability to rapidly colonise novel niches increases invasiveness. Overall, this is an impressive dataset, robust analyses (but see below), and results that are important and of interest to a wide range of ecologists.

Comments:

The intro or discussion would benefit from content on what we know about plasticity to environmental conditions in ants. For example, there is a large literature on thermal tolerance and plasticity in ants especially variation in tolerance among populations of the same species and how acclimation may allow ants to increase their thermal niche.

Line 43-45. A study that is conspicuously missing from the literature cited that is relevant here (even if just a single species approach) is: Fitzpatrick et al. 2006. The biogeography of prediction error: why does the introduced range of the fire ant over-predict its native range. *Global Ecology and Biogeography*. In particular, their finding that fire ants initially occupied areas similar in niche to native populations then expanded to different climates. Is there evidence for similar patterns with any of the species examined here? Specifically, are successfully introduced species likely to colonise novel climates or are their introductions restricted to similar niches initially (and "invasive" versus "alien" species can be compared). Also, how do the results of Fitzpatrick compare with the findings here?

Line 57-59. For some species, the native range is not very well known (or known at all). How were points confirmed as being from native populations?

Line 62-66. Summary states should be included here. Were the stats / p-values corrected for the large number of comparisons? Line 249 Statistical Analyses – does not appear that a Bonferroni was used but perhaps needed given the large number of comparisons?

How did the large variation in the number of occurrence points (309 + 552) influence climate envelope estimates? The greater number of records, the larger the variation in climate the species will occupy, and therefore the higher probability that a significant effect could be detected between native and introduced points. Many introduced species are poorly studied and therefore have few points relative to well-studied invasive species. Does the overall pattern of differences in niche breadth hold if introduced species with relatively few occurrence records (e.g. less than 50 or 100) are excluded (I appreciated that species with less than 10 occurrences were already excluded)?

Line 115-120. As a "control" - what would it look like if you sampled points from across the range of ant species that have never been introduced (that is all "native" populations) but varied in their overall range size? Would you expect widespread species to exhibit niche shifts between the two sample groups by chance while species with small ranges would not? I am curious if this pattern could result as a sampling artifact from species with large ranges have more climate variation by default.

Editorial Suggestions:

Line 25-27. First sentence could be broken into two sentences.

Line 53. The 100 worst invasive species list is not really a quantification of anything. This sentence is not necessary and can be deleted.

Reviewer #2:

Remarks to the Author:

I have submitted my review as an attachment because I have included a figure that cannot be drawn here.

The manuscript investigates whether climatic niche shifts of invasive species are larger than those of non-invasive alien species. This is a key prediction of plasticity-based hypotheses of invasiveness, which suggest that invasive species show higher plasticity under new environmental conditions than non-invasive alien species. Contrary to expectation, they found that invasive ant species showed smaller niche shifts than non-invasive alien species. They further found that species with the smallest native niches and range sizes, experienced the greatest niche shifts (independent of invasion status). This study challenges the assumption that invasive species are particularly successful in novel climates. I believe that the findings of the study are novel and will be of interest to others in the community and the wider field.

The manuscript is well written and was a pleasure to read. The methods and statistical analyses appear to be appropriate. I think that it will be suitable for publication in this journal following major revision. Below are some of my concerns and suggestions. I have also made detailed comments in the manuscript document.

I think that ants are a good taxon to use to explore this hypothesis. Below, I suggest an additional factor that may be worth mentioning in your justification for selecting ants as the study taxon. Many of the invasive species that we have today are the result of introductions for horticulture (ornamental plants and plantation forestry) and these species are usually selected for particular traits, often for hardiness. This human selection bias could be a confounding factor in studies that investigate this hypothesis. I think that it would be worth pointing out the human filter of species selection is less likely to play a role for ants because they are generally not introduced to new regions deliberately or selected for particular traits.

I have concern regarding the methods used to thin the occurrence records. In order to reduce spatial bias in the records, the dataset was thinned so that occurrence records that were closer than 0.5 km apart were excluded. The spatial resolution of the climatic predictor variables used was 2.5 minutes (about 4 km across). This means that even after thinning, several occurrences could fall within in the same climatic grid cell. I am not sure to what extent this is a problem in this dataset but I think it would have been better to have used the thinning approach and additionally to have excluded any duplicate records within the same grid cell.

An important issue that I think has been overlooked is that the time since introduction of the species. The introduction, naturalisation, invasion continuum is a key concept in invasion biology, which is conceptualised as a series of barriers (Blackburn et al. 2011). Species are moving across this continuum all the time and it can take a long time for a species to move across the continuum from introduction to invasion. Some introduced species that are currently non-invasive aliens (introduced or naturalised) may become invasive in future (although many will not). This idea is embodied in the concept of invasion debt (Essl et al. 2011; Rouget et al. 2015). What is not known from the current dataset is how long ago these species were introduced. Are many of the non-invasive aliens just recently introduced species that will become invasive given sufficient time?

A key finding of this study is that invasive species showed greater niche overlap between native and introduced ranges than non-invasive alien species. I argue that recently introduced species will initially tend to have less similar climatic niches between native and introduced ranges, but that this similarity will likely increase over time (see Fig. 1 below). I think that this may be a way to explain why invasive species tend to have greater similarity between native and introduced range realised niches. The implication is that invasive species have had sufficient time in order to expand their ranges and occupy a broader range of climatic conditions in the introduced range than non-invasive

aliens. Alternatively, non-invasive species have been introduced to areas with sub-optimal climatic conditions (but still within the fundamental niche) and have thus not (yet) become invasive.

Fig. 1 – Conceptual diagram showing that niche similarity (in two dimensions) is likely to increase from the early stages after introduction (time 1) as a species expands its geographic range to occupy a greater part of its fundamental niche (time 2). This assumes that the species does not survive outside its fundamental niche in the introduced range.

In the explanation of niche similarity between native and introduced ranges (line 109), I think what you are suggesting is that for invasive species there is greater overlap between the BAM circles (biotic, abiotic and mobility components) in the BAM diagram of Soberon and Peterson (2005). It seems that invasive species more closely approximate their case one example than non-invasive alien species? It may be helpful to cite this paper as it may be a useful way to conceptualise the differences between invasive and non-invasive alien species.

Although some important points (and words of caution) are made about the applications of species distribution models (SDMs) in the section on implications of these findings for SDMs (line 147), I don't think that the positive implications of the study have enjoyed enough attention. Although this study demonstrates that alien species tend to occupy regions in the introduced range that are quite different from the native range, this is not generally the case with invasive species. It could be argued that SDMs will tend to accurately predict the extent of the introduced range for invasive species (but not for non-invasive aliens), and this is what really matters in invasion biology (we are less concerned about aliens that don't become invasive). Perhaps it is possible to predict which introduced or naturalised species are likely to become invasive in time. The findings of this study suggest that species that show an increasing niche similarity between native range and introduced range (through range expansion with time) are more likely to become invasive. Although, perhaps by the time you know that the niches of the native and invaded ranges are similar, the species has already become widely established in the introduced range and it is too late to take action? I think that this point should be discussed briefly.

Another important point (related to the above) is that climatic matching (as opposed to species distribution modelling) has some value for risk assessment as invasive species appear to perform

best in regions that are most similar climatically to their native ranges (and not in novel climates). Climatic matching is often used in risk assessment protocols and this study suggests that it could have value for identifying potentially invasive species.

Line 145, changes in the fundamental niche could result from genetic admixtures through multiple introductions and selections of particular biotypes in the native range.

You may be interested in a recent paper by Pili et al. (2020).

I have used the term “introduced range” in preference to “non-native range” but leave the choice of terminology to you.

I thought that Table 1 with the glossary of terms was a really good idea.

The point below is just an observation and not necessarily directly relevant to this manuscript.

I think that there is another important link between range size and establishment (through propagule pressure) in the introduced range. Species with a larger geographical range size in the native range means that there will be more locations (probably cities) from which humans can transport individuals to new regions (particularly for unintentional introductions). The propagule pressure for species with larger geographical ranges is thus probably higher than those with small ranges. A larger range size in the native range likely results from a larger fundamental niche, which means that there will also be more areas in the introduced range (due to large fundamental niche) that are available for individuals of a species to be introduced to by humans and where they are likely to survive and establish. If the above is true then one would expect that invasive species would tend to have larger native range sizes than non-invasive alien species (but your study does not show this).

Reviewed by Mark Robertson – 12 August 2020

References

- Essl F, Dullinger S, Rabitsch W, Hulme PE, Hulber K, Jarosik V, Kleinbauer I, Krausmann F, Kuhn I, Nentwig W, Vila M, Genovesi P, Gherardi F, Desprez-Loustau M, Roques A, Pysek P (2011) Socioeconomic legacy yields an invasion debt. *PNAS* 108: 203-207. doi: 10.1073/pnas.1011728108. doi: 10.1073/pnas.1011728108
- Rouget M, Robertson MP, Wilson JR, Hui C, Richardson DM (2015) Invasion debt - quantifying future biological invasions. *Diversity and Distributions*. 22: 445-456. doi: 10.1111/ddi.12408
- Soberon J, and Peterson AT (2005) Interpretation of models of fundamental ecological niches and species' distributional areas. *Biodiversity Informatics*, 2, 1-10.
- Pili AN, Tingley R, Sy EY, Diesmos MLL, Diesmos AC (2020) Niche shifts and environmental non-equilibrium undermine the usefulness of ecological niche models for invasion risk assessments. *Scientific Reports* 10: 7972 doi: 10.1038/s41598-020-64568-2.
- Blackburn TM, Pysek P, Bacher S, Carlton JT, Duncan RP, Jarošík V, Wilson JR, Richardson DM (2011) A proposed unified framework for biological invasions. *Trends in Ecology and Evolution* 26: 333-339. doi: 10.1016/j.tree.2011.03.023

Reviewer 1

The authors ask if invasive species' success stems from being able to colonise novel climates. Specifically, they test if non-native populations of invasive species exhibit greater climate variation relative to native populations, than alien species (that is, species that have successfully introduced but are not considered "invasive"). They find that introduced population of "invasive" ants are, on average, closer to the climate range of native populations than non-invasive alien ants. This finding challenges the prediction that increased plasticity or ability to rapidly colonise novel niches increases invasiveness. Overall, this is an impressive dataset, robust analyses (but see below), and results that are important and of interest to a wide range of ecologists.

Comments:

The intro or discussion would benefit from content on what we know about plasticity to environmental conditions in ants. For example, there is a large literature on thermal tolerance and plasticity in ants especially variation in tolerance among populations of the same species and how acclimation may allow ants to increase their thermal niche.

We thank the reviewer for his/her excellent suggestion and have included a section about thermal tolerance and plasticity in ants, which may explain why so many ant species shift their niche.

Line 43-45. A study that is conspicuously missing from the literature cited that is relevant here (even if just a single species approach) is: Fitzpatrick et al. 2006. The biogeography of prediction error: why does the introduced range of the fire ant over-predict its native range. *Global Ecology and Biogeography*. In particular, their finding that fire ants initially occupied areas similar in niche to native populations then expanded to different climates. Is there evidence for similar patterns with any of the species examined here? Specifically, are successfully introduced species likely to colonise novel climates or are their introductions restricted to similar niches initially (and "invasive" versus "alien" species can be compared). Also, how do the results of Fitzpatrick compare with the findings here?

We agree with the reviewer that this study is highly relevant and have cited it in the introduction (see line 52-54). *Solenopsis invicta*, the species studied by Fitzpatrick et al., (2006) also showed high expansion (39%) in our study. Their findings that the species has first colonized areas similar to their native niche, followed by a later niche expansion, is intriguing. We fully agree with the reviewer that the niche dynamics over time deserve further investigation. It would be exciting to test if species start generally by colonizing climatically similar regions and subsequently move to more distinct climates. We have added this idea to the discussion. Testing this hypothesis is unfortunately not possible for the 82 species in our dataset, as it would require dated invasion records for all of these species, which are not available.

Line 57-59. For some species, the native range is not very well known (or known at all). How were points confirmed as being from native populations?

We agree with the reviewer's comment. We have clarified in method section of the manuscript that we have used the 'native' and 'non-native' categories determined by the database AntMaps (<https://antmaps.org/about.html>), where ant experts have compiled distribution data using published literature. Ideally, population genetics could be used to refine the native/alien range distinctions, but such data is available only for a minority of alien ant species.

Line 62-66. Summary states should be included here. Were the stats / p-values corrected for the large number of comparisons? Line 249 Statistical Analyses – does not appear that a Bonferroni was used but perhaps needed given the large number of comparisons?

We agree with the reviewer. We have now corrected the p-values using a Benjamini–Hochberg correction, and have added this to the statistical analysis section of our methods. This did not change the number of species with significant equivalency tests.

How did the large variation in the number of occurrence points (309 + 552) influence climate envelope estimates? The greater number of records, the larger the variation in climate the species will occupy, and therefore the higher probability that a significant effect could be detected between native and introduced points. Many introduced species are poorly studied and therefore have few points relative to well-studied invasive species. Does the overall pattern of differences in niche breadth hold if introduced species with relatively few occurrence records (e.g. less than 50 or 100) are excluded (I appreciated that species with less than 10 occurrences were already excluded)?

We agree with the reviewer, this is a really interesting point, and should be considered when comparing occurrence-based models between species. We have addressed this point in the Supplementary Material:

‘We tested if the number of occurrence points had an effect on the results. To do this, we tested if the test metrics D overlap and Expansion changed with the number of native occurrence points (after thinning- see Methods). We found no correlation between D overlap and the number of native occurrence points (Kendall's rank correlation tau, $R\tau=-0.04$, $p=0.63$). There was also no difference in the number of occurrence points among species belonging to different expansion groups (10-100%), (Kruskal-Wallis test, $X^2= 2.65$, $p\text{-value} = 0.10$).’

We have now added this point in the main text.

Line 115-120. As a “control” - what would it look like if you sampled points from across the range of ant species that have never been introduced (that is all “native” populations) but varied in their overall range size? Would you expect widespread species to exhibit niche shifts between the two sample groups by chance while species with small ranges would not? I and curious if this pattern could result as a sampling artifact from species with large ranges have more climate variation by default.

We agree with the reviewer that the effect of large ranges having more climate variation is interesting. In our study, we found that species with larger native ranges (and niches) showed smaller shifts – they essentially occupy similar niches in the non-native range. Perhaps, across all species of ants, those with larger distributions also occupy larger niches. However, if this is true, we would not be able to conclude that this is due to an “artifact” of sampling occurrences across a larger space, based on computational methods only. Indeed, large distributions and larger realized niches may be the result of larger physiological tolerances and reflect the size of the species’ fundamental niche. The referee has brought up an important fundamental question, which we hope will be addressed in future studies using experimental methods.

Editorial Suggestions:

Line 25-27. First sentence could be broken into two sentences.

Line 53. The 100 worst invasive species list is not really a quantification of anything. This sentence is not necessary and can be deleted.

We agree with the reviewer's editorial suggestions, and have made the appropriate edits.

Reviewer 2

The manuscript investigates whether climatic niche shifts of invasive species are larger than those of non-invasive alien species. This is a key prediction of plasticity-based hypotheses of invasiveness, which suggest that invasive species show higher plasticity under new environmental conditions than non-invasive alien species. Contrary to expectation, they found that invasive ant species showed smaller niche shifts than non-invasive alien species. They further found that species with the smallest native niches and range sizes, experienced the greatest niche shifts (independent of invasion status). This study challenges the assumption that invasive species are particularly successful in novel climates. I believe that the findings of the study are novel and will be of interest to others in the community and the wider field.

Comments:

The manuscript is well written and was a pleasure to read. The methods and statistical analyses appear to be appropriate. I think that it will be suitable for publication in this journal following major revision. Below are some of my concerns and suggestions. I have also made detailed comments in the manuscript document.

We thank the reviewer for his/her comments! We appreciated very much the detailed edits and suggestion on the word document, which improved greatly the readability of our manuscript.

I think that ants are a good taxon to use to explore this hypothesis. Below, I suggest an additional factor that may be worth mentioning in your justification for selecting ants as the study taxon. Many of the invasive species that we have today are the result of introductions for horticulture (ornamental plants and plantation forestry) and these species are usually selected for particular traits, often for hardiness. This human selection bias could be a confounding factor in studies that investigate this hypothesis. I think that it would be worth pointing out the human filter of species selection is less likely to play a role for ants because they are generally not introduced to new regions deliberately or selected for particular traits.

We agree with the reviewer, this is a good point! We have included this information about ants in the introduction, see line 49-51.

I have concern regarding the methods used to thin the occurrence records. In order to reduce spatial bias in the records, the dataset was thinned so that occurrence records that were closer than 0.5 km apart were excluded. The spatial resolution of the climatic predictor variables used was 2.5 minutes (about 4 km across). This means that even after thinning, several occurrences could fall within in the same climatic grid cell. I am not sure to what extent this is a problem in this dataset but I think it would have been better to have used the thinning approach and additionally to have excluded any duplicate records within the same grid cell.

The reviewer brings up an important point about spatial thinning and the resolution of data. Spatial thinning is important for different reasons apart from grid cell resolution, for example ensuring that the sampled ants are not from the same colony. To address the point raised by the reviewer, we quantified the number of grid cells with multiple occurrence points. Only 12 (out of 82) species had more than 2 occurrence points in a single grid cell (see table below showing the number of grid cells that have more than 2 occurrences for

each species). Even though this makes only a small difference to our dataset, we re-analysed the data based on the reviewer's suggestion to keep the thinning distance, but also remove any occurrences which were in the same grid cell. This caused only extremely minor differences in test metrics and p-values (see result section) and did not change our results and conclusions.

No. native grid cells >2 occurrences	No. non-native grid cells >2 occurrences	Species
3	0	Anochetus mayri
2	0	Lasius niger
0	5	Linepithema humile
0	1	Monomorium floricola
0	6	Paratrechina longicornis
0	2	Pheidole megacephala
5	0	Solenopsis geminata
0	2	Solenopsis invicta
0	3	Tapinoma melanocephalum
0	1	Tetramorium delagoense
0	2	Trichomyrmex destructor
1	3	Wasmannia auropunctata

An important issue that I think has been overlooked is that the time since introduction of the species. The introduction, naturalisation, invasion continuum is a key concept in invasion biology, which is conceptualised as a series of barriers (Blackburn et al. 2011). Species are moving across this continuum all the time and it can take a long time for a species to move across the continuum from introduction to invasion. Some introduced species that are currently non-invasive aliens (introduced or naturalised) may become invasive in future (although many will not). This idea is embodied in the concept of invasion debt (Essl et al. 2011; Rouget et al. 2015). What is not known from the current dataset is how long ago these species were introduced. Are many of the non-invasive aliens just recently introduced species that will become invasive given sufficient time?

We agree with the reviewer and think that this is a really interesting point, as the time since invasion may affect the niche dynamics. We acknowledge that subset alien species becomes invasive over time – and this may concern some of the species that are currently classified as alien species. However, we think that this would only have a small influence on the results, given that most alien ant species have been introduced for a long time and have started colonizing new areas even before World War II (Bertelsmeier et al. 2017, *Nat Ecol Evol*). Therefore, most species currently recorded as “alien” have had sufficient time to become invasive. We have added this point to the discussion.

A key finding of this study is that invasive species showed greater niche overlap between native and introduced ranges than non-invasive alien species. I argue that recently introduced species will initially tend to have less similar climatic niches between native and introduced ranges, but that this similarity will likely increase over time (see Fig. 1 below). I think that this may be a way to explain why invasive species tend to have greater similarity between native and introduced range realised niches. The implication is that invasive species have had sufficient time in order to expand their ranges and occupy a broader range of climatic conditions in the introduced range than non-invasive aliens. Alternatively, non-

invasive species have been introduced to areas with sub-optimal climatic conditions (but still within the fundamental niche) and have thus not (yet) become invasive.

Following the previous point, we agree with the reviewer it would be interesting to test if species generally start by colonizing areas with a similar climate to their native areas. To do that, one would need precise invasion histories with dated occurrences for all 82 species, which is unfortunately not available. Contrary to the expectations outlined by the referee, a previous study on the fire ant (*Solenopsis invicta*) (Fitzpatrick et al., 2007) found the species tended to first invade areas more similar to the native range. Over time, the species expanded its niche, colonizing areas with a climate outside of its native niche. This contradicts the idea that species which have been invasive longer would have more similar niches to their native range. However, we think that this is an intriguing hypothesis, which we have added to the discussion, hoping that it will stimulate future work on niche dynamics over time of species with known invasion histories.

In the explanation of niche similarity between native and introduced ranges (line 109), I think what you are suggesting is that for invasive species there is greater overlap between the BAM circles (biotic, abiotic and mobility components) in the BAM diagram of Soberon and Peterson (2005). It seems that invasive species more closely approximate their case one example than non-invasive alien species? It may be helpful to cite this paper as it may be a useful way to conceptualize the differences between invasive and non-invasive alien species.

We agree with the reviewer and thank him/her for highlighting this key reference to add to the text. The BAM model is a great way to visualize our point, and thus we have added a paragraph, placing our interpretation within the BAM framework.

Although some important points (and words of caution) are made about the applications of species distribution models (SDMs) in the section on implications of these findings for SDMs (line 147), I don't think that the positive implications of the study have enjoyed enough attention. Although this study demonstrates that alien species tend to occupy regions in the introduced range that are quite different from the native range, this is not generally the case with invasive species. It could be argued that SDMs will tend to accurately predict the extent of the introduced range for invasive species (but not for non-invasive aliens), and this is what really matters in invasion biology (we are less concerned about aliens that don't become invasive). Perhaps it is possible to predict which introduced or naturalised species are likely to become invasive in time. The findings of this study suggest that species that show an increasing niche similarity between native range and introduced range (through range expansion with time) are more likely to become invasive. Although, perhaps by the time you know that the niches of the native and invaded ranges are similar, the species has already become widely established in the introduced range and it is too late to take action? I think that this point should be discussed briefly.

We agree with the reviewer and think this is a really great point. For invasion biologists, it is encouraging that the most invasive species showed the smallest niche shifts, indicating that SDM predictions for these species may be the most reliable. The referee is correct that modelling species distributions based on their current occurrences will yield better predictions for invasive than alien species – even if we don't know yet which ones will be the worst invaders in the future. But probably, researchers and managers care more about generating good predictions for species that actually cause problems. We thank the referee for this interesting thought and have added it to the discussion.

Another important point (related to the above) is that climatic matching (as opposed to species distribution modelling) has some value for risk assessment as invasive species

appear to perform best in regions that are most similar climatically to their native ranges (and not in novel climates). Climatic matching is often used in risk assessment protocols and this study suggests that it could have value for identifying potentially invasive species.

We agree with the reviewer. We had not previously considered the use of niche shifts on risk assessment studies, so this is a great addition to highlight the relevance of this paper! We have included this point in combination with the previous point.

Line 145, changes in the fundamental niche could result from genetic admixtures through multiple introductions and selections of particular biotypes in the native range.

We agree with the reviewer that this is a possibility, and have mentioned this.

You may be interested in a recent paper by Pili et al. (2020).

We thank you for bringing this paper to our attention. This is a very interesting paper on niche shifts, particularly their use of multiple methods to assess niche shifts. We have included it as a reference.

I have used the term “introduced range” in preference to “non-native range” but leave the choice of terminology to you.

We have kept ‘non-native’ range but acknowledge that these terms are interchangeable

I thought that Table 1 with the glossary of terms was a really good idea.

Thank you!

The point below is just an observation and not necessarily directly relevant to this manuscript.

I think that there is another important link between range size and establishment (through propagule pressure) in the introduced range. Species with a larger geographical range size in the native range means that there will be more locations (probably cities) from which humans can transport individuals to new regions (particularly for unintentional introductions). The propagule pressure for species with larger geographical ranges is thus probably higher than those with small ranges. A larger range size in the native range likely results from a larger fundamental niche, which means that there will also be more areas in the introduced range (due to large fundamental niche) that are available for individuals of a species to be introduced to by humans and where they are likely to survive and establish. If the above is true then one would expect that invasive species would tend to have larger native range sizes than non-invasive alien species (but your study does not show this).

We think this is a really interesting point. Larger native range sizes are likely to result in higher probability to become introduced somewhere else due to higher probability of coming into contact with humans which can disperse them. In our study, we found that species that expanded their niche the least were those with large range sizes and niche sizes in their native range (see figure 3). However, we found no difference in the size of the native range between alien and invasive species (see below).

Comments by reviewer 2 directly on manuscript:

These comments were really helpful and we have made all the suggested modifications.

How is it possible to have almost complete niche overlap and low expansion but at the same time have significantly divergent niches? This seems counter-intuitive given what you have just said in the previous sentence. Some clarification of this point would be helpful.

We agree that this sentence was misleading and have changed “almost complete overlap” to “high overlap” (the highest D overlap we observed was 0.8). It is indeed possible to have high niche overlap, low expansion and a significantly different niche. These are three distinct metrics of niche shifts – the latter is measuring the overall shift in the density of occurrences along a PCA axis. To illustrate the subtleties of the different metrics, we have provided the example of *Nylanderia bourbonica*, which had a high niche overlap ($D=0.66$) and an expansion of only 2%. However, as illustrated by the figure below, the density of occurrences has shifted between the native (blue) and invaded range (red). Therefore, random permutations of the native occurrences include niche space which is absent in the non-native range. As a result, an equivalency test indicates a significantly divergent niche due to the high density of non-native occurrences towards the left-hand side of the PCA-axis. We have made this clearer now in the main text.

Nylanderia bourbonica

Line 141: Although the approach used in this study cannot quantify the fundamental niche. This can only be done using physiological experiments.

We agree and discuss it this in the following paragraph.

Line 151: However, on average they will be more reliable for the species that matter: those that will become invasive.

We agree and have added this argument to the discussion

Line 166: Not sure that I follow this

We have re-written this sentence to make it clearer.

References

Bertelsmeier, C., Ollier, S., Liebhold, A. & Keller, L. Recent human history governs global ant invasion dynamics. *Nat. Ecol. Evol.* **1**, 0184 (2017).

Blackburn TM, Pysek P, Bacher S, Carlton JT, Duncan RP, Jarošík V, Wilson JRU,

Essl F, Dullinger S, Rabitsch W, Hulme PE, Hulber K, Jarosik V, Kleinbauer I, Krausmann F, Kuhn I, Nentwig W, Vila M, Genovesi P, Gherardi F, Desprez-Loustau M, Roques A, Pysek P (2011) Socioeconomic legacy yields an invasion debt. *PNAS* 108: 203-207. doi: 10.1073/pnas.1011728108. doi: 10.1073/pnas.1011728108

Fitzpatrick, M. C., Weltzin, J. F., Sanders, N. J. & Dunn, R. R. The biogeography of prediction error: why does the introduced range of the fire ant over-predict its native range? *Glob. Ecol. Biogeogr.* **16**, 24–33 (2007).

Pili AN, Tingley R, Sy EY, Diesmos MLL, Diesmos AC (2020) Niche shifts and environmental nonequilibrium undermine the usefulness of ecological niche models for invasion risk assessments. *Scientific Reports* 10: 7972 doi: 10.1038/s41598-020-64568-2.

Richardson DM (2011) A proposed unified framework for biological invasions. *Trends in Ecology and Evolution* 26: 333-339. doi: 10.1016/j.tree.2011.03.023

Rouget M, Robertson MP, Wilson JRU Hui C, Richardson DM (2015) Invasion debt - quantifying future biological invasions. *Diversity and Distributions*. 22: 445-456. doi: 10.1111/ddi.12408

Soberon J, and Peterson AT (2005) Interpretation of models of fundamental ecological niches and species' distributional areas. *Biodiversity Informatics*, 2, 1-10.

Reviewers' Comments:

Reviewer #1:

Remarks to the Author:

The authors did a thorough job addressing my comments. I have no further suggestions.

Reviewer #2:

Remarks to the Author:

I am satisfied that the concerns raised in the review were adequately addressed.